# Achieving Resolution-Agnostic DNN-based Image Watermarking: A Novel Perspective of Implicit Neural Representation

## ABSTRACT

DNN-based watermarking methods are rapidly developing and delivering impressive performances. Recent advances achieve resolution-agnostic image watermarking by reducing the variant resolution watermarking problem to a fixed resolution watermarking problem. However, such a reduction process can potentially introduce artifacts and low robustness. To address this issue, we propose the first, to the best of our knowledge, **R**esolution-**A**gnostic **I**mage Wa-ter**M**arking (**RAIMark**) framework by watermarking the implicit neural representation (INR) of image. Unlike previous methods, our method does not rely on the previous reduction process by directly watermarking the continuous signal instead of image pixels, thus achieving resolution-agnostic watermarking. Precisely, given an arbitrary-resolution image, we fit an INR for the target image. As a continuous signal, such an INR can be sampled to obtain images with variant resolutions. Then, we quickly fine-tune the fitted INR to get a watermarked INR conditioned on a binary secret message. A pre-trained watermark decoder extracts the hidden message from any sampled images with arbitrary resolutions. By directly watermarking INR, we achieve resolution-agnostic watermarking with increased robustness. Extensive experiments show that our method outperforms previous methods with significant improvements: averagely improved bit accuracy by 7%∼29%. Notably, we observe that previous methods are vulnerable to at least one watermarking attack (*e.g.* JPEG, crop, resize), while ours are robust against all watermarking attacks.

## KEYWORDS

Resolution-agnostic; Robust blind watermarking; Implicit neural representation

## 1 INTRODUCTION

Invisible digital watermarking [35] is a technology for safeguarding intellectual property in multimedia [2, 13, 14]. Early research focused on directly modifying pixel values, wherein the lowest bit was altered to watermark images [5]. To enhance the robustness against various attacks, transformations were employed to conceal data in the frequency domain [34]. Although these traditional methods can watermark images of different resolutions,

*ACM MM, 2024, Melbourne, Australia*
© 2024 Copyright held by the owner/author(s). Publication rights licensed to ACM.
ACM ISBN 978-x-xxxx-xxxx-x/YY/MM
https://doi.org/10.1145/nnnnnnn.nnnnnnn

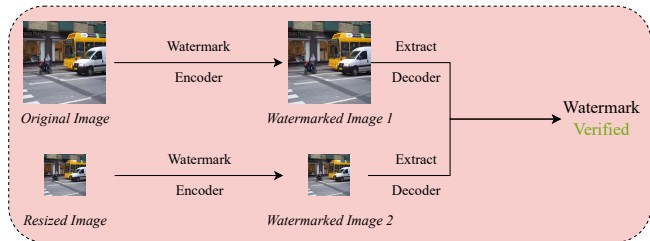

**(a) The end-to-end framework (Previous work).**

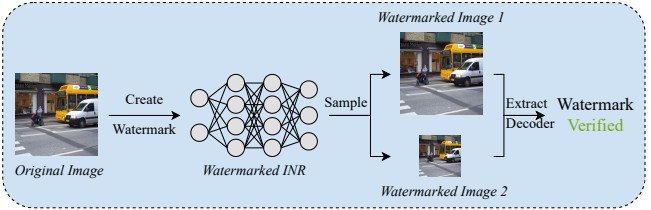

**(b) Our proposed framework RAIMark.**

Figure 1: Differences between our framework RAIMark and the previous framework. Figure 1a: The end-to-end watermarking frameworks need to re-watermark images even with a change in resolution, and fixed-resolution watermarking frameworks need to re-train models to watermark different resolution images. Figure 1b: Our framework watermarks INR and samples it to obtain watermarked images of different resolutions.

they rely on analyzing hand-crafted image features for designing watermarking techniques. With the continuous advancement of deep learning, researchers have discovered that DNN-based watermarking methods exhibit remarkable effectiveness in analyzing image features, consequently enhancing their robustness [15, 18, 19, 46]. In these DNN-based approaches, the watermark message requires expansion for subsequent interactions with images. While HiDDeN [46] and TSDL [18] directly duplicate the watermark information, increasing redundancy but lacking error correction capabilities, resulting in suboptimal robustness. To enhance the robustness, MBRS [15] incorporates a message processor module to augment error correction capabilities, thereby improving robustness. However, the message processor module constrains the image resolution that the model can watermark, *i.e.*, the entire framework has to be retrained before it can be applied to watermark images with different resolutions.

A recent work DWSF [12] tackles the above issue by reducing the variant resolution watermarking problem to a fixed resolution watermarking problem (referred to as a reduction process) by leveraging block selection. DWSF randomly selects fixed-size blocks,

which are further embedded with the secret message. The watermarked areas are first identified during extraction, and then the embedded message from these blocks is extracted. However, the bit accuracy drops fast once the extraction identifies a false watermarked block position or the watermarked block is cropped. Our experiment in Section 5.4 confirms this. Besides, we also observe artifacts of watermarked images generated by DWSF (Figure 2). **Recognizing these limitations of the DWSF's reduction process, we take a step further to ask: can we watermark images with arbitrary resolutions without relying on such a reduction process?**

To address this issue, we propose a novel perspective of watermarking images in function space; namely, we watermark the image's implicit neural representation (INR). In this paper, we propose our **R**esolution-**A**gnostic **I**mage Water**M**arking (**RAIMark**) method, which is the first INR-based watermarking approach. An INR, as a continuous representation of the image, outputs corresponding pixel values based on the given coordinates. By watermarking the INR, we can generate watermarked images of different resolutions through sampling. The watermark information is adaptively distributed across these images and can be verified with our watermark decoder, effectively addressing the limitations of previous frameworks. Furthermore, unlike previous approaches, which watermark multiple times for multiple resolutions, we only need to watermark once to get watermarked INR, and images with arbitrary resolution can be obtained by sampling from the watermarked INR, as depicted in Figure 1, which significantly reduces computational overhead in image transmissions.

RAIMark comprises three key stages, as shown in Figure 3. In stage 1, we generate the implicit neural representation by fitting it with an arbitrary-resolution image. Stage 2 involves pre-training a decoder that is independent of the image and capable of extracting watermarks from images of any resolution. In stage 3, we first generate a pre-defined message. Then, we embed the watermarks into the INR by fine-tuning the model using the pre-trained decoder to ensure the same message can be obtained from images sampled by the same INR. We obtain watermarked images of different resolutions during testing by feeding different parameters to the sampler. Our contributions are summarized as follows:

- We propose the first, to the best of our knowledge, robust, invisible, and resolution-agnostic watermarking framework RAIMark to protect images based on the implicit neural representation (INR).
- Our method leverages the watermarking of INR, enabling the generation of different resolutions of the same image directly through sampling, eliminating the need for multiple watermarking processes, and reducing computational time and costs.
- The versatility of INR as a representation for various signals, such as images, videos, and 3D models, opens up new possibilities in multimedia watermarking. This paper provides a novel perspective of watermarking INR, offering potential applications in other domains of multimedia watermarking.
- We conduct extensive experiments to demonstrate the superior performance of our method compared to state-of-the-art approaches, particularly in scenarios involving images

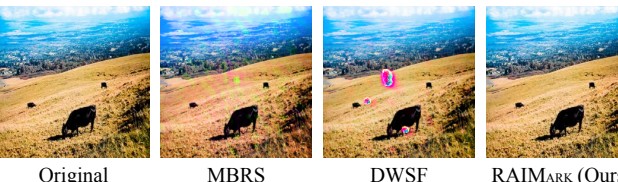

| Original | MBRS | DWSF | RAIMark (Ours) |

**Figure 2: Watermarked images of robust models. There are apparent artifacts of watermarks in the MBRS and DWSF, making it easy to recognize whether or not an image has been watermarked.**

of different resolutions. Additionally, our method exhibits enhanced resistance against both non-geometric and geometric attacks.

## 2 RELATED WORK

### 2.1 Implicit Neural Representation

Unlike explicit representations, which require explicit equations or rules to describe the object or function, implicit representations leverage neural networks to learn a mapping between inputs and outputs. In the context of computer vision, implicit neural representations (INRs) are often used in 3D shape modeling [20], scene reconstruction [28], and semantic segmentation [16]. The neural network takes a point in the space as input and produces a scalar value as output.

Implicit neural representation represents continuous signals parameterized by multi-layer perceptrons (MLPs). Early work used activation functions like ReLU and Tanh, common in machine learning [1, 22, 23, 26]. Unlike the signed distance function (SDF) in 3D space, where INR represents a continuous distance function, the pixel points of the image are discrete. If we use INR to represent an image, there are high and low-frequency regions. These activation functions are not effectiveeffective enough. Thus, periodic nonlinearities are introduced into the INR. SIREN [27] pioneeringly applied sine transform to the input coordinates, enabling INRs to fit complicated signals, which solves the problem that traditional activation functions cannot simultaneously accommodate both high and low-frequency features.

### 2.2 Image Watermarking

***Traditional watermarking.*** As a powerful means of copyright protection, digital watermarking becomes a popular area of research in real-world scenarios [7, 10, 21, 30, 31, 39]. Initial studies focused on direct changes to the pixel values of images in the spatial domain, such as the least significant bit (LSB) [6, 36]. Although LSB can achieve high invisibility with small changes in pixel values, its robustness against noises is weak. To solve this problem, researchers focused on the transformation domains, such as the Discrete Cosine Transform (DCT) [3, 25], the Discrete Fourier Transform (DFT) [29, 32], and the Discrete Wavelet Transformation (DWT) [9, 38].

***DNN-based watermarking.*** Since it is difficult for traditional methods to resist different attacks comprehensively, DNN-based watermarking emerged as computing power has increased dramatically.

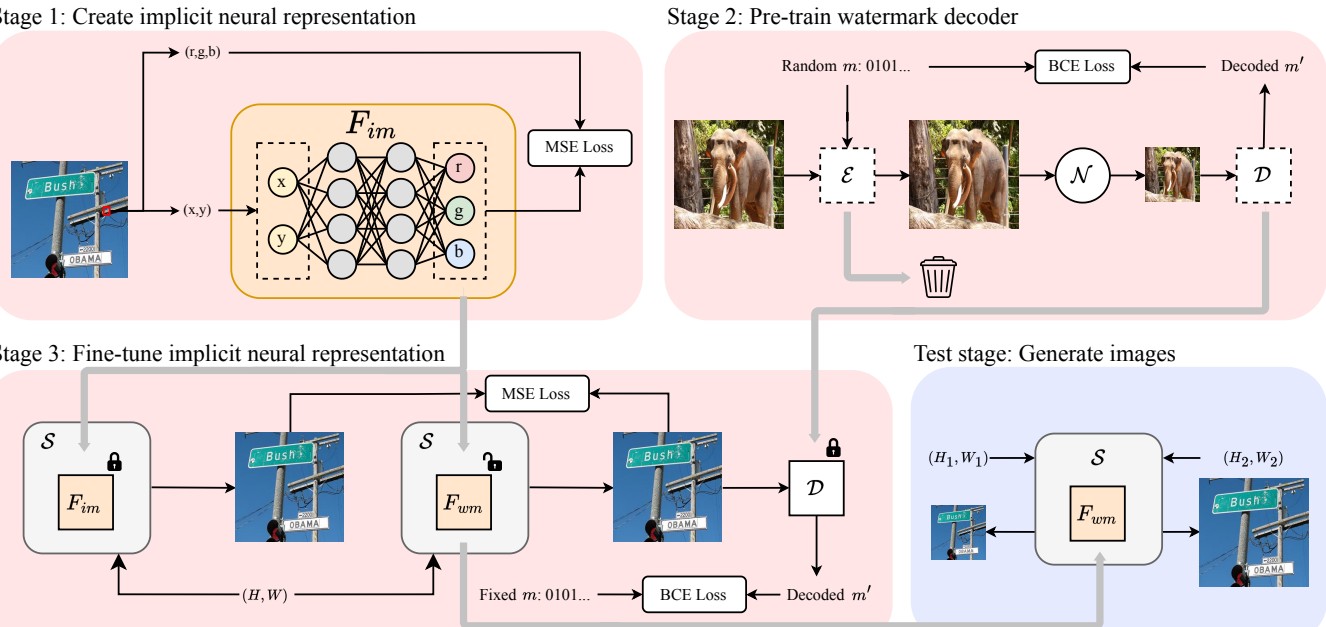

**Figure 3: Framework overview. In Stage 1, we create the implicit neural representation (INR); in Stage 2, we pre-train an end-to-end watermarking structure, then we discard the encoder and keep only the decoder; in Stage 3, we fine-tune the INR to obtain watermarked INR. In the test stage, we sample images of different resolutions using sampler $\mathcal{S}$.**

Zhu *et al.* [46] proposed HiDDeN for image watermarking, the first end-to-end DNN-based image watermarking framework. Liu *et al.* [18] proposed a Two-stage Separable Deep Learning (TSDL) framework that solved the gradient transfer problems in non-differentiable noise. MBRS [15] utilized the mini-batch strategy and combined real and simulated JPEG compression in training. Ma *et al.* [19] first incorporate an Invertible Neural Network (INN) into an embedding process, achieving excellent invisibility and robustness performance.

However, the above method gradually adds a condition to the watermark: the image resolution is fixed. This makes these watermarking methods poorly generalizable in resolution. Facing the problem of different image resolutions in real issues, Guo *et al.* [12] proposed DWSF based on selecting blocks so that a fixed-resolution watermarking framework can be used for images of other resolutions. Bui *et al.* [4] proposed a scaling-based watermarking approach. They first watermarked the images based on a specific resolution to get the residuals of the watermark. Then, the residuals are scaled and summed to the original image, thus obtaining the watermarked image.

***Generative model watermarking.*** In the case of watermarking images produced by generative models, some works processed watermarking the training set on which the model is trained [41]. To prevent multiple instances of watermarking on the generative model, some work went closer to model watermarking, merging the watermarking process and the generation process [42, 45]. The watermarking process is carried out throughout the model training process using these methods. They also have the same problem as the previous approach, and training the model is highly time

and arithmetic-intensive. The stable signature [11] showed that a quick fine-tuning of the latent decoder part of the generative model can achieve a good watermarking performance. Their work gave a good scheme for watermarking in models. It is not limited to generative models but can also be used in other models, such as the implicit neural representation.

## 3 PRELIMINARIES

### 3.1 Implicit Neural Representation

Implicit neural representation can be used as a continuous representation of an image. We can define the function $F_{im} : \mathbb{R}^2 \mapsto \mathbb{R}^3$, which maps a two-dimensional index $(x, y)$ to a three-dimensional pixel value $(r, g, b)$. A handy function $F_{im}$ uses fully-connected networks with the formulation:

$$F_{im} = \mathbf{W}_n(f_{n-1} \circ f_{n-2} \circ \cdots \circ f_1)(\mathbf{x}) + \mathbf{b}_n$$
$$f_i(x_i) = \phi(\mathbf{W}_i x_i + \mathbf{b}_i), \tag{1}$$

where $\mathbf{W}_i$ and $\mathbf{b}_i$ are weight and bias matrix of the $i$-th networks and $\phi$ is the non-linear activation function between networks. $\phi$ can be ReLU [23], Tanh [1] or sinusoidal activation function used in SIREN [27]. SIREN can better handle image details thanks to the smoothness of the sinusoidal function.

### 3.2 Sampler

We define a sampler $\mathcal{S}_{(H,W)}$, which samples INR into a $H \times W$ image. For the height side, there are $H$ indexes. For the width side, there are $W$ indexes. Combining height and width coordinates, we

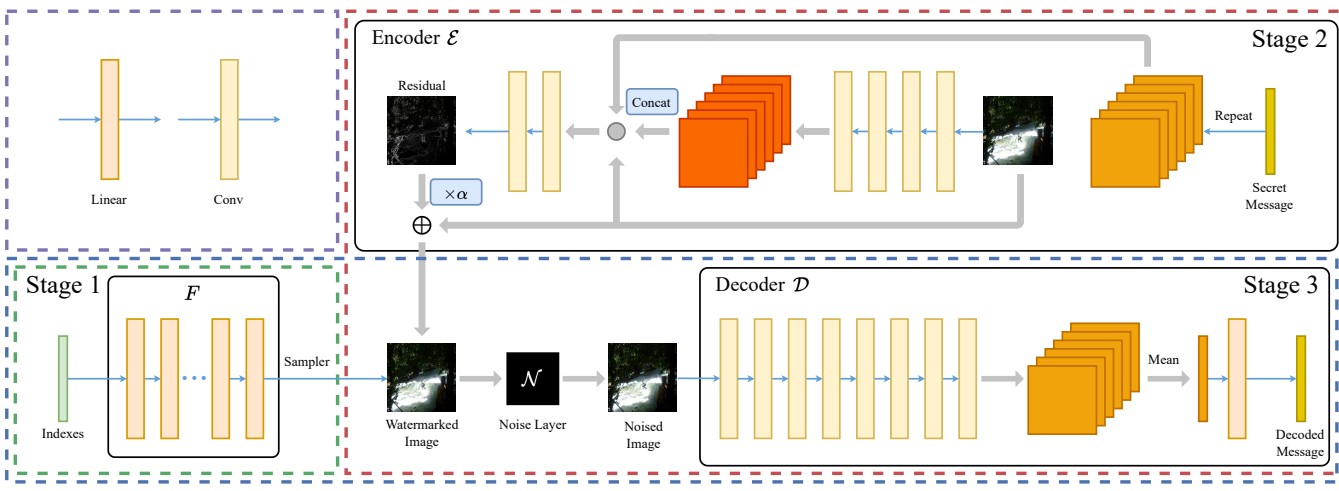

**Figure 4: Model overview. Stage 1 creates an MLP-based INR $F_{im}$ ($F$ stands for $F_{im}$ in Stage 1 and $F_{wm}$ in Stage 3) to fit the original image. Stage 2 pre-trains a decoder $\mathcal{D}$ in a DNN-based framework. Stage 3 fine-tunes $F_{im}$ with the pre-trained decoder $\mathcal{D}$ to get watermarked INR $F_{wm}$. When fine-tuning, $F_{wm}$ randomly generates images of different resolutions by changing the input parameters of the sampler, and the noise layer $\mathcal{N}$ randomly chooses an attack and applies it to the watermarked image.**

get $H \times W$ indexes:

$$(x, y) = (\frac{2 \cdot i}{H} - 1, \frac{2 \cdot j}{W} - 1), \tag{2}$$

where $i = 0, 1, \ldots, H - 1$ and $j = 0, 1, \ldots, W - 1$. The sampler contains width and height indexes uniformly distributed in the range $[-1, 1)$.

We input all indexes $(x, y)$ and get corresponding $(r, g, b)$ values. Finally, we get the sampled image by filling the image with the corresponding RGB values. Based on the continuous function property of INR, we can get the corresponding RGB values by inputting any coordinates into INR. Therefore, for consistency, in our setup, we set the coordinates of both the height and width of the samples to $[-1, 1)$.

## 4 METHOD

In this section, we give an insight into our RAIMARK, a resolution-agnostic blind image watermarking framework. Figure 4 shows the architecture of three stages in RAIMARK. Unlike the end-to-end watermarking approach, our framework embeds the watermark into the INR. No matter what resolution images are sampled from the model, these images come with their watermarks. Our framework is divided into three stages. First, we create the implicit neural representation of a given image $F_{im}$. Then, we pre-train the watermark decoder $\mathcal{D}$. Finally, we fine-tune $F_{im}$ to get the watermarked function space image $F_{wm}$, such that all images sampled from $F_{wm}$ have a given secret message through $\mathcal{D}$.

### 4.1 Creating the Implicit Neural Representation

In this stage, we choose the sine function as the activation function of the INR. The structure of INR is introduced in Section 3.1, and we initialize each sine neuron's weights before training. We set $w_i \sim \mathcal{U}(-\sqrt{6/n}, \sqrt{6/n})$, where $n$ is the number of inputs of the

neuron and $\mathcal{U}$ means uniform distribution, which ensures that the input of each sine activation is Gauss distributed with a standard deviation of 1. Specifically, for the first layer of $F_{im}$, combined with the periodicity of the sine function, we expect the output of the first neuron to span over multiple periods. Thus, we set the weight distribution of the first layer as $w \sim \mathcal{U}(-w_0/n, w_0/n)$ and set $w_0 = 30$.

Afterward, we create the INR by following the previous conditions. We define the height and width of the given image $I_o$ as $H$ and $W$. In the data processing part, to satisfy the requirements of the sampler, the first thing to do is to normalize index data into range $[-1, 1)$, which means for any index $(h, w)$, the transformation is:

$$(h, w) \rightarrow (\frac{2 \cdot h}{H} - 1, \frac{2 \cdot w}{W} - 1). \tag{3}$$

The horizontal and vertical pixel distribution density is related to $H$ and $W$. Then, after transformation, we assume, at a certain index $(x, y)$, the RGB value of the original image is $(r_o, g_o, b_o)$, which is the ground truth value. $F_{im}$ receives the same index $(x, y)$ as input and outputs the corresponding RGB value $(r, g, b)$. We must minimize the difference between $(r, g, b)$ and $(r_o, g_o, b_o)$ for a single pixel. Then, we apply a sampler $\mathcal{S}_{(H,W)}$ on $F_{im}$ and recover the image $I_F$ that is predicted by $F_{im}$. To make the predicted image similar to the original image, the loss function applies mean squared error (MSE) on $I_F$ and $I_o$:

$$\mathcal{L} = MSE(I_F, I_o) = MSE(\mathcal{S}_{(H,W)}(F_{im}), I_o). \tag{4}$$

### 4.2 Pre-training the Watermark Decoder

We first train a DNN-based watermarking framework. It optimizes both watermark encoder $\mathcal{E}$ and watermark decoder $\mathcal{D}$ to embed $n$-bit messages into images and extract them. The framework is robust against different image noises, and the decoder can receive input for any image resolution. In our framework, after training a

**Table 1: Description of geometric and non-geometric attacks.**

| Type | Attacks | Description |
| --- | --- | --- |
| Non-Geometric | $GN(\sigma)$ | Apply gaussian noise on watermarked image with standard deviation $\sigma$. |
| | $MF(k_s)$ | Blur the watermarked image by median filter with kernel size $k_s$. |
| | $JPEG(Q)$ | Compress the watermarked image with quaility factor $Q$. |
| Geometric | $Crop(s)$ | Randomly crop the $H \times W$ watermarked image with a region $(\sqrt{s} \cdot H) \times (\sqrt{s} \cdot W)$. |
| | $Resize(p)$ | Scale the $H \times W$ watermarked image into $(p \cdot H) \times (p \cdot W)$. |

robust decoder, we discard watermark encoder $\mathcal{E}$ and keep watermark decoder $\mathcal{D}$ for fine-tuning.

Formally, $\mathcal{E}$ receives a cover image $I_o \in \mathbb{R}^{3 \times H \times W}$ and an $n$-bit message $M \in \{0,1\}^n$. $\mathcal{E}$ outputs a residual image $I_r$ which is the same resolution as $I_o$. Then, we add a strength factor $\alpha$ when creating the watermarked image $I_w = I_o + \alpha \cdot I_r$, which controls the encoding strength. After applying different noises onto the watermarked image, we get the noised image $I_n = \mathcal{N}(I_w)$. $\mathcal{D}$ extracts an $n$-bit message $m = \mathcal{D}(I_w)$. The final message $M' = sgn(m)$ is the sign of $m$. We can calculate the accuracy by comparing $M$ and $M'$. The loss function we choose is Binary Cross Entropy (BCE) between the original message $M$ and the extracted message $m$:

$$\mathcal{L} = -\sum_{i=1}^{n} M_i \cdot \log(\sigma(m_i)) + (1 - M_i) \cdot \log(1 - \sigma(m_i)), \quad (5)$$

where $\sigma$ is the Sigmoid activation function, $M_i$ and $m_i$ are the $i$-th bit of original and decoded message.

Since we discard $\mathcal{E}$ afterward, the watermark invisibility is not considered in this stage. Thus, in the loss function, we consider the differencce between the original and decoded message and ignore the difference between $I_o$ and $I_w$.

## 4.3 Fine-tuning the Implicit Neural Representation

Watermarking INR is different from the end-to-end approach. In this approach, there is not a watermark embedding process. For each image generated, it comes directly from the sampling of INR. Here, we need to handle the $F_{im}$ created in the previous step for watermarking. We define the fine-tuned INR as $F_{wm}$ to distinguish it from the clean INR $F_{im}$. We fine-tune $F_{wm}$ such that the image sampled from $F_{wm}$ contains a specific message $m$ that can be extracted by $\mathcal{D}$. In the fine-tuning process, because in practice, we have many clean INRs to fine-tune, we lock all the parameters of $\mathcal{D}$ so that all fine-tuned INRs can extract the correct message from the same decoder $\mathcal{D}$.

First, we generate a pre-defined message $m = (m_1, \ldots, m_n) \in \{0,1\}^n$ for a given INR $F_{im}$. We need to save this message and use the same message during validation and testing. Then, we feed the fine-tuning INR $F_{wm}$ to a sampler $\mathcal{S}_{(H,W)}$ that outputs an image $I_s \in \mathbb{R}^{3 \times H \times W}$. Moreover, $I_s$ is the image with watermarks. During training, we change the resolution of $H$ and $W$ and add samples of different resolutions to improve generalization. The noise layers distort the sampled image $I_n = \mathcal{N}(I_s)$, and the pre-trained decoder extracts a message $m' = \mathcal{D}(I_n)$. The loss function of the message part is the BCE between extracted message $m'$ and pre-defined message $m$:

$$\mathcal{L}_{msg} = BCE(\sigma(m'), m) = BCE(\sigma(\mathcal{D}(I_n)), m). \quad (6)$$

Another objective is to improve the invisibility between $I_s$ and $I_o$. Here, since the resolution of our original image is determined, we need to generate clean images of the corresponding resolution when comparing other resolutions. We adopt $I_o = \mathcal{S}_{(H,W)}(F_{im})$ for comparison. The loss function of the image part is the MSE between $I_s$ and $I_o$:

$$\mathcal{L}_{img} = MSE(I_s, I_o)$$
$$= MSE(\mathcal{S}_{(H,W)}(F_{wm}), \mathcal{S}_{(H,W)}(F_{im})). \quad (7)$$

Thus, $F_{wm}$ is optimized by minimizing the total loss $\mathcal{L}$. We add coefficients $\lambda_{msg}$ and $\lambda_{img}$ for both parts of losses:

$$\mathcal{L} = \lambda_{msg} \mathcal{L}_{msg} + \lambda_{img} \mathcal{L}_{img}. \quad (8)$$

## 4.4 Noise Layers

Since watermarked images tend to suffer from various distortions in real-life scenarios, we added a noise layer in the training process to enhance the robustness of our model. The details of all noises we used in our method are shown in Table 1. We classify the noise into differentiable or non-differentiable depending on its realization. Differentiable noise means that after applying themselves to watermarked images, we can typically perform the reverse process when training. Moreover, with non-differentiable noises like JPEG compression, the backward propagation fails to produce the corresponding gradient because of the saving and reading of the images. Here, we choose Forward ASL [44], which is compatible with non-differentiable noises. Forward ASL first calculates the difference between noised image $I_n$ and watermarked image $I_w$, $I_{diff} = I_n - I_w$. Here, $I_{diff}$ has no gradient. The new noised image $I_n' = I_w + I_{diff}$ is the input of the decoder. The non-differentiable noise does not participate in the gradient propagation during the backward process. Therefore, the gradient can be back-propagated through the noise layer.

## 5 EXPERIMENTS

In this section, we conduct experiments on the effectiveness and robustness of our proposed framework RAIMark. First, we introduce the experimental settings of our process. Then, we show our results in two aspects. First, We validate the performance of our proposed method on the same test dataset and the exact resolution from two perspectives: invisibility and robustness. In terms of robustness, we test it against non-geometric attacks and geometric attacks. Another aspect is that we demonstrate our method's

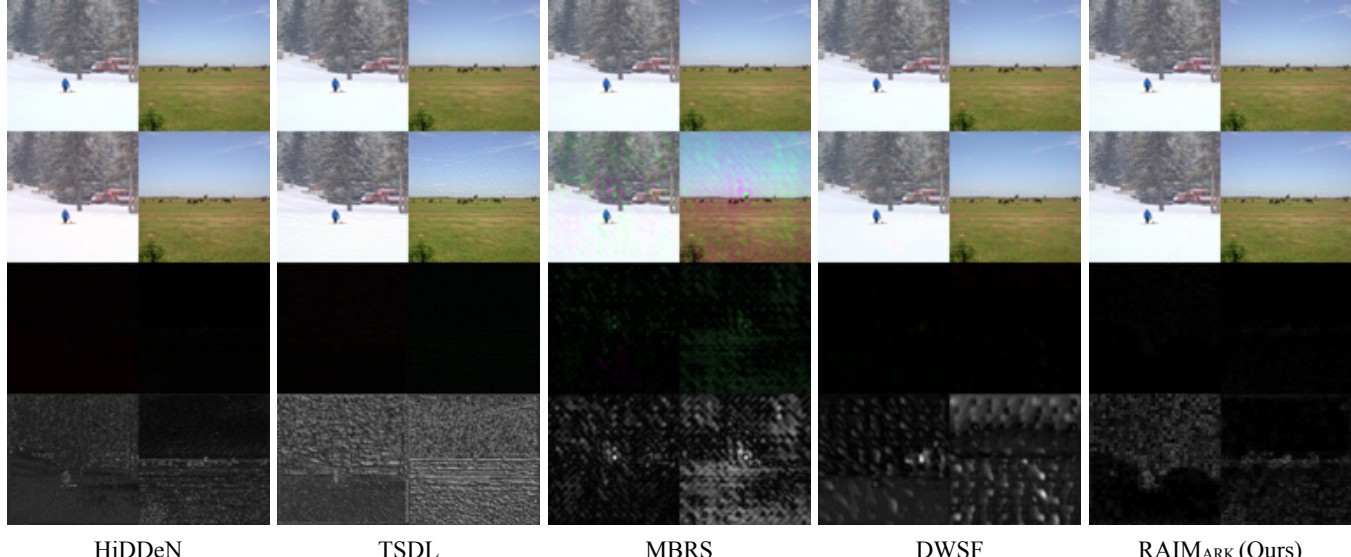

|  | HiDDeN | TSDL | MBRS | DWSF | RAIMᴀʀᴋ (Ours) |

**Figure 5: Comparison of visual quality. First row: original image $I_o$. Second row: watermarked image $I_w$. Third row: residual image $I_r$. Fourth row: normalized residual image $I_m$. We randomly choose two images in the test dataset to compare the invisibility of the watermarked images between the five methods.**

generalization by showcasing the watermark's invisibility and robustness at different resolutions. We show the generalizability of our method by testing three chosen resolutions in a fine-tuned process and three other resolutions commonly used on the screen. We choose the resolutions used during our fine-tuning process and three commonly used resolutions on screens for testing.

### 5.1 Implementation Details

Our RAIMᴀʀᴋ chooses COCO [17] as the dataset for all three phases. PyTorch implements the framework [24] and executes on Ubuntu 22.04 with an Intel Xeon Gold 5318Y CPU and an NVIDIA A100 GPU.

When creating INR, we use images whose resolution is 256×256 to fit the implicit neural representation. We choose Adam optimizer with a learning rate of $1 \times 10^{-4}$. We train $F_{im}$ for about 5000 epochs. The difference between the two images, $I_F$ and $I_o$, is invisible to the naked eye.

We select 10000 images from the COCO dataset in the decoder pre-training process. The input image resolution is set to 256 × 256. We set the message length to 30 to maintain consistency with the subsequent fine-tuning. The optimizer is Lamb [40] with learing rate of $1 \times 10^{-2}$. We choose CosineLRScheduler [37] to schedule the learning rate, which decays to $1 \times 10^{-6}$. This process is done in 500 epochs.

In the fine-tuning process, to ensure invisibility and robustness over different resolutions, we fine-tune the INR in three samples, $256 \times 256$, $384 \times 384$, and $512 \times 512$. We utilize Adam optimizer with a learning rate of $5 \times 10^{-5}$. We fine-tune the INR under non-geometric attacks and geometric attacks. Our choice for the coefficients $\lambda_{msg}$ and $\lambda_{img}$ are $5 \times 10^5$ and $3 \times 10^3$. We fine-tune 500 epochs and choose the best-performing model as the watermarked INR $F_{wm}$.

### 5.2 Metrics

The two main indicators for our watermarking model are robustness and invisibility. For different watermarked INR $F_{wm}$ in testing, robustness is measured by the accuracy between pre-defined message $m$ and the extracted message $m'$. We can get the *Accuracy* (%) by calculating the bit error rate (BER):

$$Accuracy = 1 - BER$$
$$= (1 - \frac{1}{n} \times \sum_{i=1}^{n}(m_i \oplus m_i')) \times 100\%. \tag{9}$$

where $\oplus$ is the exclusive or operation between bits.

For the invisibility, we measure the item peak signal-to-noise ratio (PSNR). We suppose $I_o$ and $I_w$ are original images and watermarked images.

$$PSNR(I_o, I_w) = 10 \times \log_{10} \frac{MAX_I^2}{MSE(I_o, I_w)}, \tag{10}$$

where $MAX_I$ is the maximum possible pixel value of the image and $MSE$ is the mean squared error.

### 5.3 Baseline

Our baseline for comparison are [46], [18], [15] and [12]. All these methods are DNN-based watermarking frameworks, and their authors open-source their code. [46], [18] and [12] can extract message from image of any resolution. We can train their author's open-source code directly. For [15], MBRS can only accept fixed-resolution input images. So, when the watermarked image is distorted by cropping or resizing, we need to scale it to its original resolution.

**Table 2: Comparison with SOTA methods. We train models with combined noise layers. We also test them with the same test dataset. PSNR is measured for RGB channels, and robustness is measured by bit accuracy (%).**

| Models | Invisibility | | Robustness | | | | | | | AVG |
|---|---|---|---|---|---|---|---|---|---|---|
| | PSNR | Identity() | GN(0.05) | MF(7) | Jpeg(50) | Crop(0.25) | Resize(0.5) | Resize(2.0) | | |
| HiDDeN | 35.31 | 98.97 | 98.77 | 94.80 | 60.77 | 98.53 | 98.67 | 99.13 | | 92.80 |
| TSDL | 33.69 | 90.77 | 87.53 | 58.17 | 54.30 | 86.73 | 60.03 | 58.73 | | 70.90 |
| MBRS | 27.63 | 99.23 | 97.90 | 98.97 | 97.43 | 58.80 | 98.97 | 99.27 | | 91.76 |
| DWSF | 37.45 | 99.97 | **99.97** | **100.00** | 95.83 | 51.33 | 99.97 | **100.00** | | 92.44 |
| RAIMark (Ours) | **39.61** | **100.00** | **99.97** | **100.00** | **99.97** | **99.20** | **100.00** | **100.00** | | **99.88** |

## 5.4 Comparison with Previous Methods

In this section, we compare our method with the SOTA methods, HiDDeN[46], TSDL[18], MBRS[15] and DWSF[12]. Since the input image resolution of each method and message length vary, we choose message length $n = 30$ for a fair comparison, and the image resolution is $256 \times 256$. Therefore, we train SOTA methods and test all methods in these conditions. Table 2 shows the detailed invisibility and robustness results.

*5.4.1 Visual Quality.* In this section, we focus on the watermarked images produced by the five methods and show the visual quality of watermarked images. The comparison of visual quality is shown in Figure 5. We calculate the residual image between original and watermarked images $I_r = |I_w - I_o|$. Moreover, we calculate the greyscale image $I_g$:

$$I_g = 0.299 \times I_{r_R} + 0.587 \times I_{r_G} + 0.114 \times I_{r_B}, \quad (11)$$

where $I_{r_R}$, $I_{r_G}$ and $I_{r_B}$ are red, green and blue channels of $I_r$. Then we normalize $I_g$:

$$I_m = \frac{I_g - min(I_g)}{max(I_g) - min(I_g)} \times I_{max}, \quad (12)$$

where $I_{max} = 1$ for floating-number images and $I_{max} = 255$ for uint8 images. Then, we can measure where the methods embed watermarks by $I_m$. As a result, in the $I_m$, the brighter the place, the higher the intensity of the embedded watermark, and the darker the place, the lower the intensity of the embedded watermark.

Based on the residual image $I_r$, we can see that RAIMark outperforms other methods in invisibility at a resolution of $256 \times 256$. For example, our method achieves 39.61dB PSNR, while the largest PSNR of the previous method only achieves 37.45 dB. Notably, when the block size is large, and image resolution is low, DWSF achieves relatively lower PSNR than RAIMark because DWSF adds a relatively larger perturbation within the block. In contrast, RAIMark adds a relatively more minor global perturbation.

*5.4.2 Combined Distortions.* To show that our RAIMark is resistant to various distortions simultaneously, we fine-tune our model with a random noise layer and a random sampler. We also train models of other methods using their default strategy and replace the noise layers with ours. The noise layers include GN($\sigma = 0.05$), MF($k_s = 7$), Jpeg($Q = 50$), Crop($s = 0.25$), Resize($p = 0.5$) and Resize($p = 2.0$). We choose two parameters of Resize, which are zooming in and out. When we test, we add an Identity() layer, which adds no noise to the watermarked image. As shown in Table

2, our model performs better than other models in all these distortions. HiDDeN is weak in JPEG compression. MBRS and DWSF are weak in cropping attacks. In particular, our method achieves 100% accuracy for Identity, Median Filter, and Resize distortions. However, our model also shows weakness in the cropping attack, which accounts for the special watermarking method of our framework. Our model increases the intensity of watermarking the high-frequency part and reduces the intensity of the low-frequency part.

We observe that previous works are vulnerable to at least one image attack. For example, DWSF is vulnerable to cropping attack because its bit accuracy drops to 51.33 under cropping attack. The reason is that DWSF only hides watermarks within selected blocks. Once selected blocks are cropped, the watermark cannot be verified. MBRS is also vulnerable to cropping attack because a specific region of the watermarked image only hides part of the binary message. Once the region is cropped, the part of the binary message it hides cannot be recovered from other image regions.

## 5.5 Evaluation on Varied Resolutions

In this section, we conduct experiments on different samples to test our model's invisibility and robustness and show its generalization. We also compare our model with the baseline approach to further demonstrate the advantages of our model.

*5.5.1 Fine-tuning Strategy.* In our method, we fine-tune our model under three different resolutions and six different noises. Randomly selecting from the resolutions and noises makes the convergence of the model uncertain. The convergence trend may be more towards a specific resolution or noise, dramatically affecting generalizability.

Thus, we refine the stochastic strategy for the fine-tuning process. We expect the model to converge in a more balanced way for various scenarios. Here, we cross-combine different resolutions and noises and get all $(resolution, noise)$ pairs as a set $S_0$. In each epoch, we get a clone $S_0'$ of $S_0$. When fine-tuning, we randomly choose a pair from $S_0'$ and remove the pair from $S_0'$. When the pair $S_0'$ is empty, we finish an epoch of fine-tuning.

*5.5.2 Results on invisibility and Robustness.* In this section, we analyze the invisibility and robustness of our model on six different samples: $128 \times 128$, $256 \times 256$, $512 \times 512$, $480 \times 854$ (480p), $720 \times 1280$ (720p) and $1080 \times 1920$ (1080p).

Section 5.5.3 shows the invisibility results. For robustness, we divide the attack into two categories: non-geometric attacks (Figure 6a) and geometric attacks (Figure 6b). Our method maintains a

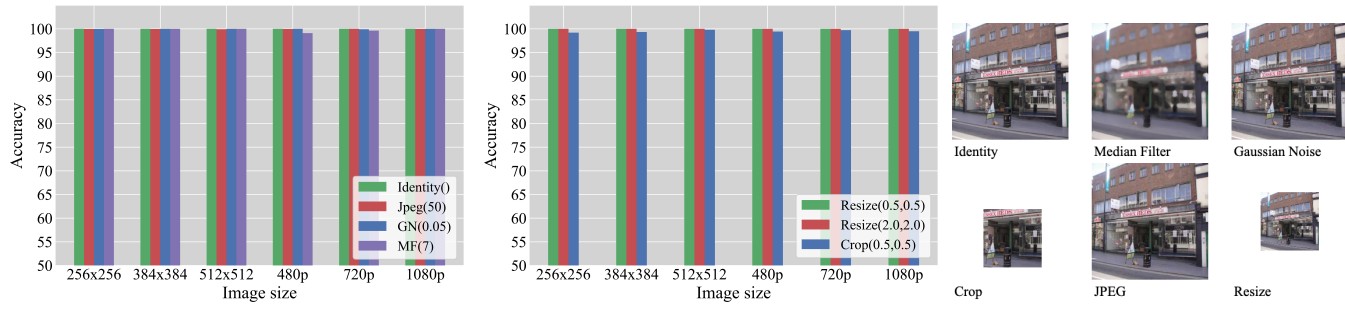

(a) The non-geometric attacks.

(b) The geometric attacks.

(c) The attack results.

Figure 6: The robustness of our model when facing non-geometric and geometric attacks. We sample $F_{wm}$ into different resolutions and apply attacks to the sampled image. We show robustness against different attacks in two categories: non-geometric and geometric attacks. Figure 6c shows the noised images after applying different attacks to the watermarked image.

Table 3: Comparison with SOTA methods in varied resolutions. HiDDeN, TSDL, DWSF and RAIMark can watermark images of any resolution. MBRS can only watermark images with the same resolution as training. "/" means the method is not applicable under the selected resolution. We measure PSNR between watermarked and original images and average bit accuracy (%) evaluated under multiple attacks.

| Models | 256x256 | | 384x384 | | 512x512 | | 480x854 | | 720x1280 | | 1080x1920 | |
|---|---|---|---|---|---|---|---|---|---|---|---|---|
| | PSNR | Acc | PSNR | Acc | PSNR | Acc | PSNR | Acc | PSNR | Acc | PSNR | Acc |
| HiDDeN | 35.31 | 92.80 | 35.50 | 92.87 | 35.80 | 92.65 | 35.97 | 92.64 | 37.14 | 92.28 | 38.28 | 92.64 |
| TSDL | 33.69 | 70.90 | 33.86 | 71.51 | 33.91 | 71.34 | 33.75 | 71.04 | 33.66 | 71.59 | 33.89 | 70.58 |
| MBRS | 27.63 | 91.76 | / | / | / | / | / | / | / | / | / | / |
| DWSF | 37.45 | 92.44 | **40.96** | 92.05 | **43.28** | 92.02 | **45.88** | 91.82 | **46.44** | 94.01 | **45.71** | 93.94 |
| RAIMark (Ours) | **39.61** | **99.88** | 39.83 | **99.90** | 39.79 | **99.96** | 39.73 | **99.79** | 39.73 | **99.90** | 39.73 | **99.92** |

high standard in all non-geometric attacks, which keeps over 99% regardless of the image resolution. In geometric attacks, zooming in and out has little effect on accuracy. The accuracy of the cropping attacks is more significant than 99%. From the results above, we observe no performance degradation when the watermarked INR is smapled to larger resolution images.

5.5.3 *Compare with Other Methods.* This section compares our method with other methods in varied resolutions. The training settings are 30-bit messages, and the image resolution is the above-mentioned six resolutions. For MBRS, their encoders utilized a message processor module, which fixed the resolution of the images, making it impossible to watermark images of other resolutions. HiDDeN and TSDL processed messages repeatedly to handle watermarking images of any resolution. DWSF achieved variant resolution watermarking through block selection.

Table 3 shows the results of five methods, in which we only compare MBRS at the trained resolution, while the other three methods are compared at all resolutions. We can observe that RAIMark and DWSF outperform previous watermarking methods. In most cases, RAIMark achieves higher bit accuracy while DWSF achieves higher PSNR. The reason is that DWSF only adds perturbation within the selected block, leaving the unselected region unchanged. However, this makes it vulnerable to cropping attacks. Once the block is falsely identified or cropped, its watermark cannot be verified, thus

DWSF has low bit accuracy. RAIMark has higher robustness when compared with DWSF because RAIMark adds a global perturbation, which makes the watermark survive various attacks. This is why RAIMark has a relatively low PSNR compared to DWSF. However, RAIMark still achieves higher than 39dB PSNR, and previous work already clearly indicated that 37dB PSNR is enough to provide good visual quality in practice[8, 33, 43].

## 6 CONCLUSION

In this paper, we have shown that reducing the variant resolution watermarking problem to the fixed resolution watermarking problem introduces artifacts and low robustness in image watermarking. To address this issue, we have proposed RAIMark to solve the variant resolution image watermarking by watermarking the implicit neural representation(INR) of the image. Different from previous methods, RAIMark does not rely on the previous reduction process by directly watermarking the continuous signal instead of image pixels. Watermarked images with arbitrary resolutions can be sampled from the watermarked implicit neural representation. Extensive experiments have demonstrated that our framework shows promising results. Overall, INR can express various multimedia resources, and our watermarking scheme provides a novel perspective to inspire subsequent resolution-agnostic watermarking frameworks.

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
