# OpenReview forum: "Achieving Resolution-Agnostic DNN-based Image Watermarking: A Novel Perspective of Implicit Neural Representation"
_acmmm.org/ACMMM/2024/Conference — MM2024 Poster_

### Official Review · Reviewer_amSG · 2024-05-22

**Rating:** 4
**Confidence:** 3

**Summary:**

This paper proposes a resolution-agnostic image watermarking (RAIMaRK) framework. The resolution-agnostic watermarking is achieved by directly watermarking the INR (implicit neural representation) of an image instead of the image pixels. The paper demonstrates the effectiveness and robustness of the proposed method through extensive experiments and comparisons with state-of-the-art methods, and the obvious improvement in performance is shown.

**Strengths:**

1. The proposed RAIMaRK framework achieves resolution-agnostic watermarking, which is not restricted by the size of images and meets the demand for watermarking images with different resolutions.
2. The paper proposes to watermark images using the implicit neural representation of images, which gives new possibilities when designing the watermarking framework.
3. The paper provides comprehensive experiments to demonstrate its superior performance compared to SOTA methods, both in terms of invisibility and robustness. The obvious improvement can be seen.

**Limitations:**

1. More evaluation results under more kinds of distortions and more parameters of distortions should be further provided to fully illustrate the robustness of the proposed method. In Table 2 of this paper, to illustrate the robustness of the proposed method, just one kind of Noise is used (GN), and just one Filter is used (MF), which is not enough to show the robustness against Noise and Filter. More kinds of Noise and Filter should be used. For example, Poisson Noise, SP Noise, Box Blur, Gaussian Filter, etc., should be also considered. Besides, the tested parameter of each distortion includes just one situation. More parameters should be considered, such as JPEG(40, 60, 70, 80), MF(3*3, 5*5), etc.
2. Ablation studies about different components of the method should be further provided to show the effectiveness of different components. For example, if the watermark decoder is not pre-trained and it is trained together in the later process, will the performance be better?
3. More examples of the output of the proposed method and comparative methods should be shown to illustrate the result of real outputs. In this paper, the shown example is limited, so it can not be well-illustrated that the watermark can be embedded without explicitly harming the visual quality of the cover image.

**Suitability:**

3

---

### Official Review · Reviewer_TYnb · 2024-05-23

**Rating:** 6
**Confidence:** 2

**Summary:**

This paper presents an innovative image watermarking method called RAIMaRK, which is based on Implicit Neural Representation to achieve resolution-agnostic image watermarking. This approach circumvents the issues of artifacts and robustness that traditional methods may face when dealing with images of varying resolutions. The paper is well-written with clear logic and thorough experimental design.

**Strengths:**

It is the first watermarking framework based on INR, offering a novel perspective on addressing image watermarking challenges. The method watermarks the continuous signal directly rather than image pixels, effectively tackling the resolution variance problem and significantly enhancing watermark robustness. The authors employ advanced deep learning techniques and validate the effectiveness of their method through extensive experiments.  Experimental results show that RAIMaRK has improved bit accuracy by an average of 7% to 29% and demonstrated excellent resistance to various watermarking attacks.

**Limitations:**

While the paper provides an overall framework and experimental results, further elaboration on some implementation details may be needed for experiment reproduction.

**Suitability:**

3

---

### Official Review · Reviewer_TTbd · 2024-05-23

**Rating:** 4
**Confidence:** 4

**Summary:**

This paper proposes a framework called resolution-agnostic image watermarking (RAIMaRK). It can watermarking based on implicit neural representation (INR). The method does not rely on the previous reduction process by directly watermarking the continuous signal instead of image pixels.

**Strengths:**

This manuscript proposes an interesting deep-learning watermarking scheme. The idea of INR watermarking has not been explored for DNN-based watermarking as far as I know. In experiments, the proposed model demonstrates good visual quality and robustness under attacks, particularly in scenarios involving images of different resolutions.

**Limitations:**

The manuscript is well structured, but the clarity of some details of the proposed method is not very good. The experimental evaluation could be further improved.  The manuscript introduces that the watermark message is fixed and pre-defined in Figure 3 and Section 4.3. Does this mean that for a fixed INR, only specific fixed messages can be watermarked? In comparison with previous methods, the original methods of HiDDeN, TSDL, MBRS, and DWSF were conducted at a resolution of 128x128, while the experiments in the manuscript selected 256x256 Will this be beneficial to the proposed method? The manuscript should consider the impact of message length on performance and discuss how the proposed approach performs in terms of computational efficiency. Is the proposal robust to severe geometric attacks (e.g. large angle rotation, padding, flipping) and hybrid attacks (e.g. rotation and scaling, cropping and rotation)? Additionally, the manuscript could also explore the limitations or potential drawbacks of the proposed method, which would provide a more balanced view of the work.

**Suitability:**

2

---

### Meta-Review · Area_Chair_QJpf · 2024-07-01

**Recommendation:** Accept (Poster)
**Confidence:** 5

**Metareview:**

The reviewers are mostly satisfied with the rebuttal and recognize that the authors contribute to the issue of resolution-independent deep image watermarking.
The manuscript is recommended to be accepted, and the authors are expected to include the responses in the rebuttal into the revised version.